# Population Health Management to identify and characterise ongoing health need for high-risk individuals shielded from COVID-19: a cross-sectional cohort study

Charlie Kenward,[1] Adrian Pratt,[2] Sam Creavin,[3] Richard Wood ID ,[2,4] Jennifer A Cooper ID [2,3]

¹NHS Bristol, North Somerset and South Gloucestershire Clinical Commissioning Group, Bristol, UK
²Department of Modelling and Analytics, NHS Bristol, North Somerset and South Gloucestershire Clinical Commissioning Group, Bristol, UK
³Department of Population Health Sciences, University of Bristol, Bristol, UK
⁴School of Management, University of Bath, Bath, UK

**Correspondence to**
Dr Charlie Kenward;
charlie.kenward@nhs.net

## ABSTRACT

**Objectives** To use Population Health Management (PHM) methods to identify and characterise individuals at high-risk of severe COVID-19 for which shielding is required, for the purposes of managing ongoing health needs and mitigating potential shielding-induced harm.

**Design** Individuals at 'high risk' of COVID-19 were identified using the published national 'Shielded Patient List' criteria. Individual-level information, including current chronic conditions, historical healthcare utilisation and demographic and socioeconomic status, was used for descriptive analyses of this group using PHM methods. Segmentation used k-prototypes cluster analysis.

**Setting** A major healthcare system in the South West of England, for which linked primary, secondary, community and mental health data are available in a system-wide dataset. The study was performed at a time considered to be relatively early in the COVID-19 pandemic in the UK.

**Participants** 1 013 940 individuals from 78 contributing general practices.

**Results** Compared with the groups considered at 'low' and 'moderate' risk (ie, eligible for the annual influenza vaccination), individuals at high risk were older (median age: 68 years (IQR: 55–77 years), cf 30 years (18–44 years) and 63 years (38–73 years), respectively), with more primary care/community contacts in the previous year (median contacts: 5 (2–10), cf 0 (0–2) and 2 (0–5)) and had a higher burden of comorbidity (median Charlson Score: 4 (3–6), cf 0 (0–0) and 2 (1–4)). Geospatial analyses revealed that 3.3% of rural and semi-rural residents were in the high-risk group compared with 2.91% of urban and inner-city residents (p<0.001). Segmentation uncovered six distinct clusters comprising the high-risk population, with key differentiation based on age and the presence of cancer, respiratory, and mental health conditions.

**Conclusions** PHM methods are useful in characterising the needs of individuals requiring shielding. Segmentation of the high-risk population identified groups with distinct characteristics that may benefit from a more tailored response from health and care providers and policy-makers.

## Strengths and limitations of this study

► Analyses are based on a linked dataset for one million individuals combining information across primary, secondary, community and mental health services.
► Criteria used for identifying high-risk individuals were subject to double clinical review and are fully documented to aid reproducibility.
► Results can facilitate health and care interventions that may be tailored for subgroups within the high-risk population.
► Results reflect the local population where the study was performed, which may limit portability to other settings.

## INTRODUCTION

SARS-CoV-2, responsible for the disease known as COVID-19, was declared a pandemic by the WHO on 11 March 2020. It has threatened to overwhelm, and in some cases has overwhelmed, health systems regardless of the economic status of the affected nation.[1 2] SARS-CoV-2 has the ability to spread quickly, partly owing to infected individuals being contagious during the early and asymptomatic phase.[3–5] While COVID-19 is a mild or flu-like illness in most people, for an estimated 14% the disease can be considered 'severe', with an estimated 6% of cases becoming 'critical' (involving life-threatening pneumonia and respiratory failure).[6] Combined with an estimated case fatality rate of approximately 1.4%,[7] this has led many countries to take drastic policy measures to suppress and contain viral transmission such as lockdown, social distancing and mask wearing, and to shield those at highest risk of severe illness through self-isolation.[8–10]

Faced with early projections that approximately 80% of the population would contract the virus over the course of the epidemic,[8] the UK Government has sought to shield the most vulnerable members of the population. In England, efforts were made to identify and inform an estimated 1.5 million people considered to be at high risk[11] of the need to self-isolate within their homes for at least 12 weeks. However, this policy comes at the potential cost of shielding-induced harm such as negative mental health effects, reduced exercise and barriers in accessing health services, which people are likely to require due to the nature of the condition(s) that has led to their shielding. Furthermore, the strategy puts a further strain on local health and care services having to meet the often complex needs of this group outside of normal ways of working. Responding to this challenge requires a detailed understanding of individuals at high risk, including a characterisation of their health and social needs and the ways in which they normally interact with health and care services.

Population Health Management (PHM) approaches can enable greater proactivity in the response of health and care services to emerging or otherwise unrecognised need, and allow for more personalised and preventative healthcare interventions.[12] As an emerging concept involving an integrated consideration of health determinants, outcomes and interventions,[13] PHM has a substantial presence in the published long term plan for England's National Health Service (NHS).[12] Yet a key challenge to embedding PHM is the ability to source and link record-level data across the health and care divide, with primary care of particular importance given the range of information relating to diagnoses, comorbidities, social status and prescriptions.[14 15] This breadth of data is required to accurately identify the high-risk individuals that require shielding (in a similar manner to its use in previous studies to identify individuals at risk of developing long-term conditions).[14 16] In addition to identifying the high-risk group, PHM approaches can also be leveraged to provide a more holistic understanding of these individuals in informing an effective response to their needs. As part of a range of descriptive analyses associated with PHM, population segmentation supports this by cutting through the complexity of large and multivariate linked datasets in determining a manageable number of population groups separable by differences in health determinants or outcomes.[17–19] This may be an important tool in understanding the composition of the high-risk group considered here.

The objective of this study was to use PHM methods to identify and characterise a high-risk population for which shielding is required, for the purposes of managing ongoing health needs and mitigating potential shielding-induced harm. Set in a large healthcare system in South West England during the early stages of the COVID-19 pandemic, this study makes use of a linked dataset containing healthcare activity and clinical, demographic and social attributes for one million individuals.

## MATERIALS AND METHODS

The REporting of studies Conducted using Observational Routinely collected Data[20] statement was used as an extension of the Strengthening the Reporting of Observational Studies in Epidemiology guidelines[21] (online supplemental material A).

### Data, application and setting

The application of this study was to the Bristol, North Somerset and South Gloucestershire (BNSSG) healthcare system, which is one-million resident health economy across a mixture of urban and rural geographies. Within the system, there are 82 General Practitioner (GP) practices and three major hospitals. This cross-sectional cohort study took place at a time understood to be early in the pandemic, shortly after the non-pharmaceutical interventions (social distancing and isolation strategies) were nationally implemented on 23 March 2020.[22]

The BNSSG system-wide dataset[23] was used to support the analysis contained in this study. This dataset forms part of the PHM infrastructure within the BNSSG healthcare system, and contains information for 1 013 940 registered individuals (from 78 contributing practices) across two tables. The first table consists of individual attributes, including the presence of clinical conditions, lifestyle factors, and demographic and socioeconomic information. These data are principally derived from GP's patient administration systems (all GP practices use EMIS Web). The second table contains information for various patient-related activities such as GP consultations, hospital admissions, mental health appointments, community visits and prescriptions. These data are sourced across primary and secondary care, mental health and community services and contains information such as specialty and relevant dates and times. A unique individual identifier is used to link the data between the tables. Eligibility criteria included all individuals registered to a contributing practice within BNSSG.

### Patient and public involvement statement

Patients and/or the public were not involved in the design, conduct, reporting or dissemination plans of this research.

### Identifying high-risk individuals

In the UK, construction of the Shielded Patient List (SPL) was led by the Chief Medical Officers (CMOs) of the four home nations, with support from the various medical Royal Colleges and Societies. From the outset, individuals were considered to require shielding if they were a member of one or more of six outlined groups. This included those who have received an organ transplant; those with specific cancers, severe respiratory conditions and rare diseases that increase the risk of infection and those who are on immunosuppression therapies or are pregnant with significant congenital heart disease.[24] In developing the specific criteria used to define membership of these groups, a set of central database searches

was first published by NHS Digital in March 2020. This accounted for approximately 900 000 individuals of the total 1.5 million people originally estimated to require shielding. In seeking to address this shortfall, the membership criteria have since evolved through additional incorporation of patient lists sent by hospital consultants and GPs, and self-reporting by the public.[11]

Given the manner in which the SPL has incrementally advanced, there are a number of issues affecting its ability to effectively identify the high-risk population for consideration within this study. First, it is exposed to variation in clinical assessment and citizen reporting, that is, there are elements of subjectivity. And second, there is anecdotal evidence emerging from front-line clinicians regarding concerns around over and under reporting of high-risk individuals contained on the SPL, that is, it is to some degree incomplete. Ultimately, this approach has not resulted in a comprehensive database aligned to the SPL for the purposes of secondary uses as required by healthcare planners.

Consequently, the high-risk group of the BNSSG population considered in this study was identified through the merger of two constituent lists. The first of these is a BNSSG-level subset of the aforementioned SPL originally released by NHS Digital,[11] accounting for approximately 900 000 individuals nationally and 14 388 individuals for the BNSSG population. This was complemented with a list of individuals as identified through a number of local searches of the BNSSG system-wide dataset created in order to match as closely as possible the definitions of the six groups as originally outlined by the CMO (identifying 24 894). Combining these lists gave 29 798 high-risk individuals. Where an exact match in search field was not possible, then it was either omitted or a proxy search term was constructed on review by two clinicians.[25] To reduce potential misclassification and information bias, high-risk criteria and coding lists were subject to double clinical review. Full criteria are documented in online supplemental material B.

### Analysing the high-risk group

The high-risk group was first analysed through comparison against the remaining population, partitioned to 'low' and 'moderate' risk groups. Moderate risk is defined by eligibility for the annual injectable influenza vaccination[11] and low, or baseline, risk is defined through membership of neither the moderate nor the high-risk groups. Descriptive analyses were performed on the basis of demographics, comorbidities, healthcare utilisation and geographical distribution. Pyramid plots were used to understand the differences in age and sex between individuals of these three groups. The high-risk group was then assessed against the other groups through a comparative analysis examining summary information for setting-level healthcare utilisation, prevalence of individual chronic conditions and non-specific mortality risk (through the Charlson score[26]) in addition to other demographic and socioeconomic variables. Geospatial analyses mapped the

proportion of high-risk individuals onto the BNSSG geography at Lower Super Output Area (LSOA) level, in order to identify any differences by rurality or deprivation (as measured by the Index of Multiple Deprivation (IMD), which ranks LSOAs from most to least deprived on a scale from 1 to 10). There were very few instances of missing data since the data were derived from GP practices and other national data (any variables with missing data accounted <1% of the data). Where required, missing data were handled through a complete cases approach.

The high-risk group was segmented using cluster analysis, which identifies similar groups within a population through the use of statistical methods to maximise the difference between groups according to a given set of features. The features considered here relate to the range of demographic, clinical and social attributes that could be of interest in gaining a high-level understanding of the clusters, and moreover that would make the clusters 'actionable' in terms of the nature of possible interventions applied to mitigate shielding-induced harm. Given the range of categorical and continuous features considered (online supplemental material C), the k-prototypes clustering method[27] was selected due to its flexibility to accommodate such mixed data types. The number of clusters was selected based on a scree plot of the total within sum of squares versus the number of clusters, the Silhouette index[28] and based on the 'identifiability' and 'actionability' of the clusters in a healthcare context[18] (online supplemental material C). Both clustering and non-clustering variables were assessed for whether they were statistically different from other clusters, an approach used by recent studies to verify results and to indicate how segments differ on each clustering variable.[29 30] First, an omnibus statistical test was applied (ANOVA - Analysis of Variance, Kruskal-Wallis test and $\chi^2$ test) to confirm differences across clusters followed by 15 pairwise tests (t-test, Mann-Whitney U test and z-test for proportions) between all other clusters. A Bonferroni adjustment was made to the significance level of 0.05 using multiple test comparisons.

## RESULTS

A total of 1 013 940 individuals (50.02% female, median age: 37 years (IQR: 21–56 years)) were included within these analyses, with 29 798 individuals (2.94%) identified as 'high risk', 32.79% as moderate risk and the remaining 67.01% as low (baseline) risk. An age and sex breakdown for individuals at these various levels of risk is provided in figure 1. Figure 1 shows that, according to the definitions used, younger and middle-aged individuals are generally at relatively low risk, with females of typical reproductive age (20–44 years) at comparatively higher risk compared with males due to the additional risk borne through pregnancy (thus warranting their inclusion for the annual injectable influenza vaccine). In contrast, high-risk individuals are mostly aged between 40 years and 85 years. Figure 2 shows that the long-term conditions used to

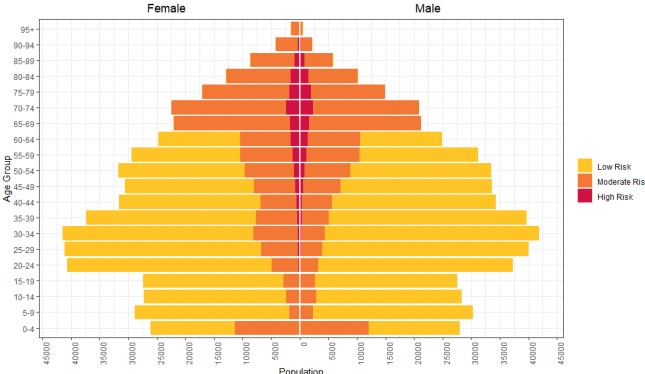

**Figure 1** Population pyramid showing absolute numbers of the population in 5-year age bands stratified by high risk (red), moderate risk (orange) and low risk (yellow).

define the 'high-risk' group (Section 'Identifying high-risk individuals') are more common with increasing age.

A comparative analysis of the low-risk, moderate-risk and high-risk groups is provided in table 1 with regard to demographic and socioeconomic information, historical healthcare utilisation and burden of comorbidity. The high-risk group, as expected through definition, is on average older, has a higher comorbidity rate (Charlson Score increases across risk stratas, from a median of 0 (IQR: 0–0) for low risk to 4 (3–6) for high risk) and has greater healthcare utilisation for all appointment types. For instance, compared with the group at low risk, the high-risk group has on average 6 times the number of primary care and community contacts over the preceding calendar year and two times the number compared with the moderate-risk group. In terms of comorbidities, the high-risk group has a much higher proportion of individuals diagnosed with cancer in the past 5 years (24.03%) and with chronic obstructive pulmonary disease (COPD) (34.84%) compared with the rest of the cohort. In addition, the proportion of those with cardiovascular conditions (19.59%) in the high-risk group is higher compared with the rest of the cohort (3.48%) and for those who

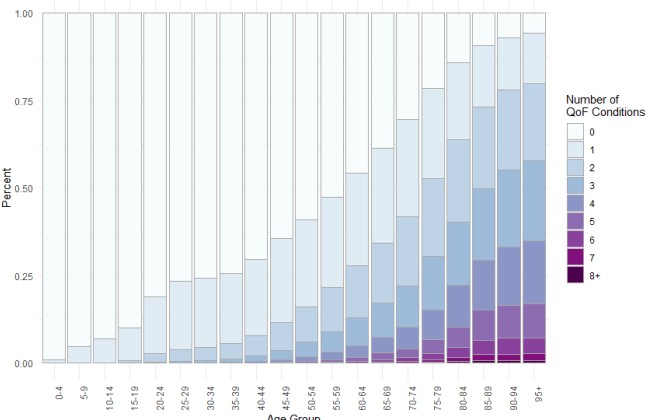

**Figure 2** Number of long-term conditions by 5-year age band, with long-term conditions assessed through the Quality and Outcomes Framework (QOF) definitions.

are on medication which requires monitoring (24.13% vs 1.52%). Further comparisons are given in online supplemental material D.

The concentration of high-risk individuals within the various LSOAs of the BNSSG healthcare system is illustrated in figure 3. Given that each LSOA contains an average of 1500 individuals, those LSOAs larger in size represent rural geographies, while those smaller in size and clustered together represent the greater population densities associated with urban conurbations. The large town of Weston-super-Mare (population 80 000) appears in the lower area of the figure and the city of Bristol (population 463 000) appears towards the centre on the eastern side. Of the six localities comprising the BNSSG system, the centre and eastern areas of Bristol have the lowest proportion at high risk (2.29%); and people in this area were younger compared with the other five localities (overall median age: 32 years, cf 38 years; 'high-risk' median age: 64 years cf 69 years). The greatest concentration of high-risk individuals (3.88%) is for the locality, including Weston-super-Mare and surrounding areas. The rural areas outside of Bristol account for 55% of the high-risk group (16 255 out of 29 798). Rural areas have a higher proportion of the population at high risk (3.33%) compared with urban city areas (2.91%) (p<0.001, test of two proportions).

Table 2 shows the results of the segmentation analysis of the high-risk group (29 798) into six subgroups. Two-hundred and twenty four individuals did not have an LSOA within the BNSSG area but were registered to a BNSSG practice and were removed from the analyses. The top users of the high-risk group were removed (0.41%) leaving a high-risk population of 29 454. Clustering variables differ significantly across the segments as expected since the algorithm inherently creates clusters to differ as much as possible based on the variables. Non-clustering variables were also significantly different across groups. Pairwise comparisons between segments identify which segments differ from others, for instance, for Cluster two the cardiovascular proportion (44.24%) differs significantly from all other clusters.

Cluster 1 contains only 0.57% of the high-risk group and accounts for those with the highest need in regard of mental health services (79% had a mental health diagnosis). This cluster had the greatest proportion of individuals with dementia (10.59%). Those with dementia in this cluster had a mean age of 78 versus 56 for those without dementia. Half the individuals with dementia had a mental health condition, whereas 17.76% of individuals without dementia had a mental health condition. Although this cluster had the highest proportion of people with dementia, membership of this group was largely defined by 78.82% of individuals having a mental health condition. Median Charlson Comorbidity Index for those with dementia was 6 (IQR: 5–7.75) and those without dementia was 3 (IQR: 2–4).

Cluster 2 has the greatest age (median age: 76 years) and most frequent utilisation of primary, secondary

**Table 1** Comparative analysis of the low-risk (n=679 457), moderate-risk (n=304 685) and high-risk (n=29 798) groups with regard to demographic and socioeconomic information, historical healthcare utilisation and burden of comorbidity

| Variable | Low risk | Moderate risk | High risk |
|---|---|---|---|
| **Demographic and socioeconomic** | | | |
| Age (median, IQR) | 30 years (18–44 years) | 63 years (38–73 years) | 68 years (55–77 years) |
| Female | 48.25% | 53.82% | 51.77% |
| Deprivation by IMD decile* (median, IQR) | 6 (3–8) | 6 (4–9) | 6 (3–8) |
| Urban/rural | | | |
| Rural town and fringe | 2.34% | 3.13% | 2.97% |
| Rural village and dispersed | 3.85% | 5.19% | 4.81% |
| Urban city and town | 93.81% | 91.68% | 92.22% |
| Local authority | | | |
| Bristol | 52.24% | 42.58% | 43.74% |
| North Somerset | 19.79% | 26.40% | 27.14% |
| South Gloucestershire | 27.97% | 31.02% | 29.12% |
| Has a carer | 0.31% | 1.46% | 2.89% |
| Housebound | 0.04% | 1.61% | 3.73% |
| **Healthcare utilisation (calendar year 2019)** | | | |
| Primary and community care contacts (median, IQR) *Mean contacts per 1000 population* | 0 (0–2) *1620* | 2 (0–5) *4753* | 5 (2–10) *9766* |
| Mental health attendances (median, IQR) *Mean contacts per 1000 population* | 0 (0–0) *226* | 0 (0–0) *432* | 0 (0–0) *622* |
| Secondary care elective consultations and admissions (median, IQR) *Mean contacts per 1000 population* | 0 (0–0) *966* | 0 (0–4) *2958* | 6 (2–14) *10 756* |
| Secondary care emergency attendances and admissions (median, IQR) *Mean contacts per 1000 population* | 0 (0–0) *295* | 0 (0–0) *527* | 0 (0–2) *1184* |
| **Comorbidities** | | | |
| Cardiovascular condition | 0.068% | 11.09% | 19.59% |
| Cancer diagnosed† | 0.29% | 1.85% | 24.03% |
| Mental health condition | 9.44% | 12.54% | 16.96% |
| Diabetes | 0.07% | 14.94% | 19.95% |
| Dementia | 0.017% | 2.31% | 2.68% |
| Asthma | 0.82% | 17.03% | 22.27% |
| COPD | 0.04% | 3.05% | 34.84% |
| **Other** | | | |
| Drugs that require monitoring‡ | 0.64% | 3.49% | 24.13% |
| Charlson Score (median, IQR) | 0 (0–0) | 2 (1–4) | 4 (3–6) |
| Smoking (Current Smoker) | 12.94% | 10.21% | 16.21% |

*For IMD, note that 1 is the most deprived decile and 10 is the least deprived decile.
†Diagnosis in past 5 years.
‡Drugs that require monitoring include immunosuppressant/immunomodulators (previous 6 months) and biologic/monoclonal medication (previous 6 months), including rituximab (previous 12 months) and other drugs requiring monitoring (previous 2 months), fully specified in online supplemental material B.
COPD, Chronic Obstructive Pulmonary Disease; IMD, Index of Multiple Deprivation.

and community care. Clinically, this segment is characterised by a high prevalence of COPD (58%), cardiovascular disease (44%), diabetes (31%) and dementia (8%). Cluster 3 has the lowest age (median 51 years) and highest asthma prevalence (76%). Cluster 4 has the highest use of elective secondary care appointments with 88% having had a diagnosis of cancer in the past 5 years. Cluster 5 is characterised by the majority (78%) requiring physiological, biochemical or pharmacological monitoring of prescribed medication. Of this cluster, 60% had

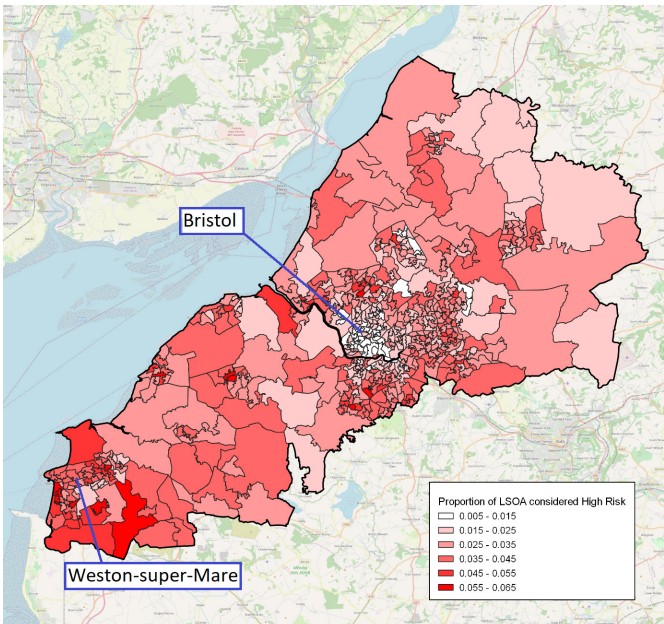

**Figure 3** Geographical map of the Bristol, North Somerset and South Gloucestershire healthcare system illustrating the concentration of high-risk individuals at Lower Super Output Areas (LSOA) level.

an immunosuppressant/immunomodulator prescribed in the previous 6 months, 22% had a biologic/monoclonal medication prescribed in the previous 6 months and 11% had another type of drug requiring monitoring in the previous 2 months (eg, lithium, warfarin and antipsychotic medications, see online supplemental material B and C). Finally, Cluster 6 is characterised by an 85% COPD prevalence and the second highest smoking prevalence (25%). This cluster is also notable from having lower than average healthcare utilisation across all considered settings.

## DISCUSSION
### Summary of main findings
In aggregated routine cross-sectional data from a single healthcare system (1 013 940 population), 2.94% (29 798) of the population met the UK NHS criteria for being at high risk of severe COVID-19 illness. The proportion of people at high risk of COVID-19 increased from 2.88% of those aged 50–54 years to 13.21% of those aged 80–84 years, whereas 9.46% of those aged 90–94 years and only 6.79% of those aged 90 years and above were at 'high risk'. Compared with those who were not at high risk of COVID-19, the high-risk group were older, more frequently attended healthcare and scored highly on the Charlson Comorbidity Index. Rural and semi-rural areas, and coastal towns had higher proportions of the population at high risk of severe COVID-19 illness than urban and inner-city areas. Segmentation of the high-risk population identified six clusters reflecting the distinct characteristics of types of individuals that may benefit from a more tailored response from health and care providers

and policy-makers. At the time of writing, this is the first study in the published literature to evaluate the attributes and needs of a shielded population through the application of PHM approaches.

### Strengths and limitations
A key strength is the large dataset combining individual level data available from the electronic medical record with granular data on healthcare utilisation, demography and geography across a population of over one million people. In particular, the dataset includes healthcare utilisation, which covers community, primary and secondary care, and mental health services. A further strength is the collaborative co-production of findings drawing on a multidisciplinary team comprising clinicians, data scientists and commissioners. This has been possible through the PHM infrastructure embedded at the healthcare system under study. Finally, we demonstrated the potential to develop a comprehensive and accurate search based on this rich, linked data. The criteria used for identifying high-risk individuals were subject to double clinical review and were fully documented to aid reproducibility.

The study had its limitations. First, the findings were derived from a single health and care system in one country (the UK), which may hinder generalisability. However, the BNSSG population closely matches the age and sex distribution for England using data from the Office for National Statistics and Public Health England.[31 32] The results may, therefore, be portable to other English systems, or countries with similar demographics and health status. Additionally, the principle that population segmentation can be rapidly used to identify actionable insights and support a commissioning response to a healthcare emergency is broadly applicable.

Second, the findings rely on routinely coded data, and do not use free text in the medical record which may introduce information bias. While reflecting clinical practice, the ascertainment of clinical characteristics (especially diseases) by coded data is likely to both overclassify and underclassify people as having disease. The data on medication and prescriptions are likely to be more completely ascertained but may not reflect medication usage. In contrast, healthcare utilisation data are likely to be most completely ascertained (although difficult to ascertain DNAs and administrative errors).

There is also the possibility for differential coding, that is, people in certain groups may be less likely to be correctly coded than others. Examples include people who are homeless, experiencing socioeconomic disadvantage or language barriers, or those in the travelling community. This may result in differential misclassification especially towards people who are at high-risk of COVID-19 illness. Finally, a computer-driven approach can identify individuals quickly from large patient lists and records, but additional benefit could be obtained by providing further support to practices to identify high risk individuals using their knowledge of individual patient health and care needs.

**Table 2** Segmentation of high-risk group (n=29 454), detailing attributes of the six subgroups as identified through cluster analysis

| Cluster | 1. Complex mental health (n=170) | 2. Older complex (n=1372) | 3. Younger asthma (n=5327) | 4. Recent cancer (n=6612) | 5. Drug monitoring (n=6892) | 6. Low utilisation COPD (n=9171) | Overall omnibus test |
|---|---|---|---|---|---|---|---|
| | | | **Clustering variables** | | | | |
| Age, mean (years) Median (IQR) | 58.4* 58 (50.25–70.00) | 72.7† 76 (66.00–84.00) | 48.3* 51 (34.00–64.00) | 65.3† 68 (57.00–76.00) | 65.1† 69 (57.00–77.00) | 71.2† 72 (64.00–80.00) | AN <0.000 |
| Female, n (%) | 105 (61.76) | 871† (63.48) | 3892† (74.32) | 2665* (40.31) | 4850† (70.37) | 2891* (31.52) | $\chi^2$ <0.000 |
| Primary and community care contacts, median (IQR) Mean contacts per 1000 population | 10* (5–20) 16 006 | 49* (25–83) 61 847 | 4 (2–9) 6721 | 4 (2–9) 6808 | 4 (2–8) 6493 | 4 (2–9) 6733 | KW <0.000 |
| Mental health attendances, median (IQR) Mean contacts per 1000 population | 45.5* (34–62) 49 876 | 0* (0–0) 612 | 0‡ (0–0) 175 | 0* (0–0) 53 | 0‡ (0–0) 155 | 0‡ (0–0) 155 | KW <0.000 |
| Secondary care elective consultations and admissions, median (IQR) Mean contacts per 1000 population | 6‡ (2–13) 10 288 | 11* (5–20) 15 136 | 5† (1–11) 7860 | 16* (8–28) 21 332 | 7† (3–13) 9020 | 3* (0–7) 4818 | KW <0.000 |
| Secondary care emergency attendances and admissions, median (IQR) Mean contacts per 1000 population | 2* (0–3) 2365 | 6* (4–9) 6820 | 0* (0–2) 1089 | 0* (0–1) 940 | 0* (0–1) 680 | 0* (0–1) 786 | KW <0.000 |
| Cardiovascular condition (current), n (%) | 27 (15.88) | 607* (44.24) | 435† (8.31) | 1018‡ (15.4) | 1093‡ (15.86) | 2569† (28.01) | $\chi^2$ <0.000 |
| Cancer diagnosed in the past 5 years, n (%) | 25† (14.71) | 230† (16.76) | 358* (6.84) | 5813† (87.92) | 224* (3.25) | 431* (4.7) | $\chi^2$ <0.000 |
| Mental health, n (%) | 134* (78.82) | 308† (22.45) | 1181† (22.55) | 766† (11.58) | 1067† (15.48) | 1516† (16.53) | $\chi^2$ <0.000 |
| Diabetes, n (%) | 39 (22.94) | 419† (30.54) | 746‡ (14.24) | 1098‡ (16.61) | 1384† (20.08) | 2168† (23.64) | $\chi^2$ <0.000 |
| Dementia, n (%) | 18† (10.59) | 112† (8.16) | 35* (0.67) | 91* (1.38) | 192* (2.79) | 329† (3.59) | $\chi^2$ <0.000 |
| Asthma, n (%) | 54† (31.76) | 322† (23.47) | 3970* (75.81) | 358* (5.41) | 507* (7.36) | 1358* (14.81) | $\chi^2$ <0.000 |
| COPD, n (%) | 78† (45.88) | 800† (58.31) | 1053* (20.11) | 277* (4.19) | 404* (5.86) | 7677* (83.71) | $\chi^2$ <0.000 |
| | | | **Non-clustering variables** | | | | |
| Drugs that require monitoring, n (%) | 67* (39.41) | 243* (17.71) | 640* (12.22) | 351† (5.31) | 5396* (78.29) | 403† (4.39) | $\chi^2$ <0.000 |
| Smoking (Current Smoker), n (%) | 63† (37.06) | 211† (15.38) | 888† (16.96) | 626† (9.47) | 679† (9.85) | 2317† (25.26) | $\chi^2$ <0.000 |

Continued

**Table 2** Continued

| Cluster | 1. Complex mental health (n=170) | 2. Older complex (n=1372) | 3. Younger asthma (n=5327) | 4. Recent cancer (n=6612) | 5. Drug monitoring (n=6892) | 6. Low utilisation COPD (n=9171) | Overall omnibus test |
|---|---|---|---|---|---|---|---|
| Urban/rural, n (%) | | | | | | | |
| Rural town and fringe (rural) | 6 (3.53) | 45 (3.28) | 125 (2.39) | 226 (3.42) | 234 (3.40) | 237 (2.58) | For Urban vs rural: $\chi^2$ <0.000 |
| Rural village and dispersed (rural) | 5 (2.94) | 41 (2.99) | 200 (3.82) | 408 (6.17) | 407 (5.91) | 359 (3.91) | |
| Urban city and town (urban) | 159 (93.53) | 1286 (93.73) | 4912 (93.79) | 5978 (90.41) | 6251 (90.70) | 8575 (93.5) | |
| IMD decile, median (IQR) | 4 (2–7) | 5 (3–8) | 5 (3–8) | 7* (4–9) | 7* (4–9) | 5 (2–8) | KW <0.000 |
| Learning disabilities and autism, n (%) | 7‡ (4.12) | 20† (1.46) | 105‡ (2.00) | 28‡ (0.42) | 32‡ (0.46) | 38‡ (0.41) | $\chi^2$ <0.000 |
| Housebound, n (%) | 19* (11.18) | 328* (23.91) | 56† (1.07) | 104† (1.57) | 181* (2.63) | 393* (4.29) | $\chi^2$ <0.000 |
| Has a carer, n (%) | 10 (5.88) | 134† (9.77) | 75‡ (1.43) | 137 (2.07) | 148 (2.15) | 341† (3.72) | $\chi^2$ <0.000 |
| Is a carer, n (%) | 3 (1.76) | 68 (4.96) | 162 (3.09) | 210 (3.18) | 271 (3.93) | 343 (3.74) | $\chi^2$=0.002 |
| Charlson Score, median (IQR) | 3† (2–5) | 6* (4–7) | 2* (1–4) | 5* (3–6) | 4† (2–5) | 5* (3–6) | KW <0.000 |

Bonferroni adjustment to significance level: 0.05/15=0.0033 (based on 15 pairwise tests).
*Significantly different from five other clusters (all other clusters).
†Significantly different from four other clusters.
‡Significantly different from three other clusters.
§Drugs that require monitoring include immunosuppressant/immunomodulators (previous 6 months), biologic/monoclonal medication (previous 6 months) including rituximab (previous 12 months) and other drugs requiring monitoring (previous 2 months), fully specified in online supplemental material B.
AN, Analysis of Variance; COPD, Chronic Obstructive Pulmonary Disease; IMD, Index of Multiple Deprivation; KW, Kruskal-Wallis test.

## Interpretation

The resulting size of the derived BNSSG high-risk population is comparable to an estimated expected number of around 27 500 (1.5 million national high-risk individuals out of 55 million population applied to our cohort). This verifies the search criteria developed to identify these high-risk individuals. Through leveraging more granular data as contained in established PHM infrastructure, this approach both promotes enhanced risk sensitivity through reducing the risk of over and under reporting (as is possible through coarser central database searches) and enables a comprehensive specification of the search criteria.

Of the six clusters, four were determined by criteria related to the definition of high risk, that is, 'younger asthma' (n=5327), 'recent cancer' (n=6612), 'drug monitoring' (n=6892) and 'low utilisation COPD' (n=9171). Four of the six clusters had a higher proportion of women (younger asthma: 74%; drug monitoring: 70%; older complex: 63% and mental health 62%) than men, whereas the COPD cluster and recent cancer had a higher proportion of men (68% and 60%). This reflects known demographics: women are more likely than men to have asthma and autoimmune disease (which often requires drug monitoring), live longer than men and are more likely to be diagnosed with mental health disorders.[33–36] In contrast, men are more often diagnosed with cancer and COPD than women.[37] An interesting finding was the small (n=170) but important complex mental health cluster, the result of a high prevalence of comorbidities.

## Implications

It is important to note that at the time of writing it is not known whether the UK shielding strategy has been effective in terms of reducing the burden of COVID-19 or overall harm for that defined population. Our findings are, therefore, relevant to healthcare planners and clinicians because they offer insights at the level of the population. There are three important implications for planners. First, the population at high risk of COVID-19 reflects a heterogeneous group of people who will require different interventions and response to mitigate the risk of shielding-induced harm. Mitigating interventions should be targeted at clusters among other important groups such as those with learning disabilities. For instance, targeted smoking cessation advice by text message for people in the low utilisation COPD group, targeted medication reviews for those in the drug monitoring group; and an individualised, proactive multidisciplinary care plan for the 0.02% of the population in the complex mental health group.

Second, our analysis shows that population segmentation can be used to highlight geographic areas of greatest need during a pandemic, drawing parallels with the long-term challenges facing rural and coastal town health highlighted by England's CMO.[38] Policy-makers and clinicians can use these findings to understand how the capacity of healthcare systems reflect the likely demand. The third important implication is that local and national policy-makers must explicitly consider the risk of systematic misclassification, even in times of pandemic. The shielding policy is based on at least two assertions: that the criteria as operationalised identify the group at highest risk of serious illness (a classification question), which is likely to currently underclassify high-risk populations, and that shielding reduces the risk of serious illness to an extent that outweighs any harms (a complex-intervention question), for which there is currently a lack of evidence. The former is now supported in part by a large UK-based study using the OpenSAFELY Platform, identifying independent risk factors for severe COVID-19 disease, which was consistent with the criteria used to determine the SPL,[39] although it is notable that older age was the strongest predictor of in-hospital death and was not an explicit factor in the criteria. The extent to which the latter assertion is true is as yet unknown, which is important because there are potential harms to individuals from both the misclassification in both directs and to society from misallocation of resources.

Algorithms and artificial intelligence are at risk of exacerbating health inequalities by systematic misclassification.[40] We found that in our population, the proportion classified as at high risk according to national criteria was similar for those aged over 95 years as in those aged 65–69 years. Given that the proportion at high risk increases monotonically until the age of 85 years, there is a concern that this may reflect systematic underclassification of risk in the oldest old of our population, especially because the proportion at low risk paradoxically increases from the age of 85 years and onward. Systematic misclassification could also partly explain the lower proportion of people at 'high risk' in our deprived populations who may have differential ascertainment of underlying disease and higher risk of transmission due to living arrangements. Our concerns about systematic misclassification are especially relevant given the high excess death rate in care facilities and in people of black and minority ethnic populations.[39 41] Broadening the high-risk criteria would, however, induce a further significant personal and societal cost, including people with a Charlson Score of 6 and greater would result in 45 000 people classified as at high risk, adding an additional ~15 000 people to the current shielded BNSSG population.

There are also further limitations with the principle of shielding. First, shielding has a very limited evidence base, and, therefore, as with much of the policy response to COVID-19, the international implementation has been varied. For example, South Korea did not implement shielding and has a very low number of deaths reported from COVID-19. However, at this stage, we believe that limitations in the comparability of the death rate (or indeed excess deaths) between international states preclude the comparison of death rates by interventions on a state-by-state basis. Second, shielding can be difficult to implement on a practical level for the individual. Further work could explore the strategies that individuals used to operationalise shielding on an everyday basis, the extent to which this adhered to national guidelines and whether there were any barriers to implementation that could have been better

mitigated by a more comprehensive response from the health system. However, regardless of these limitations, our report demonstrates how PHM methods can be applied at pace, during a pandemic, to identify geographic areas and characterise clusters of people who likely have additional health needs and support actionable insights to safeguard population health.

## Future work

Two initial priorities for further work are to apply these clusters to other geographic settings to determine whether the clusters hold in other healthcare settings and to prospectively determine the actual risk of severe COVID-19 illness and subsequent outcomes in people who are determined to be at high risk. With planned developments of our dataset, a further study could examine in greater depth the observed risks and outcomes in terms of hospitalisation, discharges to care settings and mortality among the shielded population. The high-risk definition was developed at pace and with the limited evidence, and expert opinion available at the time. However, risk factors that were not included were age, ethnicity, male sex, multimorbidity or other single comorbidities such as diabetes or heart disease, each of which has been shown to be independently associated with increased risk of severe disease[7 39 42–45]

As we move into a new phase of the pandemic,[46] which may involve further case surges, the development of accurate, risk prediction models are going to be an essential part of local health and care responses.[47] In addition, research is needed to understand and maximise the classification accuracy of search criteria used for population level datasets, as has been performed in the clinical practice research database,[48] and how these can be integrated as support mechanisms for front-line clinicians, who ultimately hold responsibility for making decisions on who meets the criteria for shielding. Critically, it needs to be established whether shielding is effective. Shielding may be undermined by non-concordance with the rules, a reliance on social, medical and care interactions, or from being resident in a high-risk setting such as a care home. This is especially important given the lack of evidence for the rest of the non-shielded population having acquired 'herd immunity'. Without such evidence, there is a risk to the well-being of current and future shielded populations, induced by the shielding process. Finally, it should be examined whether there are more effective strategies such as widespread use of face coverings in public and effective 'track and trace' systems.

## CONCLUSIONS

PHM methods applied to a system-wide linked dataset are useful in identifying and characterising a population at high risk of COVID-19 for which shielding is required. Cluster analysis of the high-risk group revealed heterogeneous groups that may benefit from a more tailored response from health and care providers and policy-makers, and prompts further examination of a policy which is not without potential harm.

**Acknowledgements** Support in developing and providing the searches of primary care data was provided by Rhys Lewis and Dr Jacob Lee, OneCare. Geographical distribution map of COVID-19 high-risk proportions was produced by Abbie Reynolds-Beer, Bristol North Somerset and South Gloucestershire Clinical Commissioning Group.

**Contributors** CK designed the study, constructed the high-risk criteria using the Bristol North Somerset and South Gloucestershire Clinical Commissioning Group's system-wide dataset and drafted and revised the paper. JAC designed the study, carried out data analysis, including cluster analysis modelling, and drafted and revised the paper. RW contributed to study design and data analytics, and drafted and revised the paper. SC constructed the high-risk criteria, provided clinical input and interpretation and reviewed the paper. AP contributed to data analysis, construction of the dataset/variables and drafted the manuscript. All authors have approved the final version of the manuscript for publication.

**Funding** This study was supported by the NIHR Bristol Biomedical Research Centre at University Hospitals Bristol, and Weston NHS Foundation Trust and the University of Bristol.

**Map disclaimer** The depiction of boundaries on this map does not imply the expression of any opinion, whatsoever on the part of BMJ (or any member of its group) concerning the legal status of any country, territory, jurisdiction or area or of its authorities. This map is provided without any warranty of any kind, either express or implied.

**Competing interests** None declared.

**Patient and public involvement** Patients and/or the public were not involved in the design, or conduct, or reporting, or dissemination plans of this research.

**Patient consent for publication** Not required.

**Ethics approval** Yorkshire & The Humber—Leeds East Research Ethics Committee granted NHS HRA Research Database approval for the Bristol, North Somerset and South Gloucestershire (BNSSG) system-wide database in 2020 (REC Reference Number: 20/YH/0185, date: 28 July 2020). Ethical approval is given for processing of personal data by the research database team for research conducted on COVID-19-related priority areas identified by the wider health and care system (in accordance with the Health Service Control of Patient Information Regulations 2002 (COPI) directive for COVID-19-related analyses). Under General Data Protection Regulation (GDPR), the data controller was BNSSG Clinical Commissioning Group. The data from general practice electronic medical records were extracted according to existing data sharing agreements and in line with GDPR and Caldicott principles under pandemic conditions and in accordance with government guidelines and the COPI directive.

**Provenance and peer review** Not commissioned; externally peer reviewed.

**Data availability statement** No data are available. Technical appendix and coding lists available in online supplemental files. The R code and data variables used for analyses are included on the following GitHub page (https://github.com/nhs-bnssg-analytics/kprotocluster). Local data used for the study are not publicly available.

**ORCID iDs**
Richard Wood http://orcid.org/0000-0002-3476-395X
Jennifer A Cooper http://orcid.org/0000-0001-9364-7353

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
