## [Reviewer comments · BMJ Open]

ARTICLE DETAILS

TITLE (PROVISIONAL)	Population Health Management to identify and characterise ongoing health need for high-risk individuals shielded from COVID-19: A Cross Sectional Cohort Study
AUTHORS	Kenward, Charlie; Pratt, Adrian; Creavin, Sam T; Wood, Richard; Cooper, Jennifer

VERSION 1 – REVIEW

REVIEWER	Elizabeth Price Great Western Hospital Swindon SN3 6BB
REVIEW RETURNED	22-Jun-2020

GENERAL COMMENTS	This interesting paper describes the use of Population Health management methods to identify and characterise individuals requiring shielding and their health care needs The paper describes the processes used to identify the individuals and then cohorts them into 6 clusters. They have identified 6 distinct cohorts. It is interesting to note that 4 of the 6 had a high proportion of females and 2 of the 6 had a median age below 60 years. They discuss the fact that the shielding criteria failed to take in to account factors such as age, ethnicity and gender that have since emerged as predictors of poor outcome. What would be really interesting would be to know how many of these patients developed COVID-19 and what their outcomes were. Is this possible? Knowing which of the 6 clusters was actually at highest risk would allow further targeting of resources and inform future shielding decisions. Minor corrections: Page 3, line 40 – missing end bracket
--

REVIEWER	Mohamamd S Razai St George's University of London
REVIEW RETURNED	28-Jun-2020

GENERAL COMMENTS	Thank you for the opportunity to review. I think the study is well done from a methodological point of view but I have very strong reservations about the study's clinical utility or purpose. I am pleased that the authors have discussed some of the problems in the limitations of the study. PHM methods involve a lot of work and effort as the authors can tell. The authors have done a good job of analyzing a huge amount of
--

data which is commendable. I have briefly summarised my issues with the study below.

1. The limitations show that PHM is not a reliable way to identify high risk individuals. There are problems with information bias - incorrect/incomplete/inadequate coding, relying on error-prone prescriptions datasets etc. These limitations clearly show the problems of the approach adopted in the UK. Identifying people through PMH is probably better than the way it was done by the NHS England but this is still a centralised computer generated list. Many clinicians in primary care including myself found the classification of most of our 'highly vulnerable' patients incorrect, those who were supposed to 'shield' did not receive any information but many patients were incorrectly classified as very high risk based on outdated or incorrect data held centrally. A centralised, top down approach of identifying high risk individuals is ineffective and certainly leads to confusion, frustration, misallocation of resources and potential harm. A better approach is supporting and providing resources to practices and community care facilities to identify high risk individuals using their knowledge of individual patients under their care and their health needs.

2. As the authors acknowledge there is absolutely no evidence that 'shielding' works at all. The problem is the assumptions about 'shielding' which was a political strategy adopted in the UK, Sweden and some other Western countries. It is not a scientific strategy. 'Shielding' is based on 2 erroneous assumptions. 1) If only a small proportion of the population 10-15% are very highly vulnerable to COVID-19 (and economically not productive), they could be easily isolated from 90% of the population who will likely have a mild illness and will develop 'herd immunity'. The Chief Scientific Advisor on 12 March 2020 said: "Our aim is not to stop everyone getting it, you can't do that. And it's not desirable, because you want to get some immunity in the population. We need to have immunity to protect ourselves from this in the future." The underlying objective was when initially announced, to 'cocoon' those 'most at risk' whilst leaving the economy open outside, in the hope that those who survive will develop 'herd immunity'. There is no evidence that those infected develop immunity and it quickly became apparent that the strategy would lead to hundreds of thousands of deaths. A nationwide lockdown was introduced shortly after. Furthermore, evidence from China shows that 80% of infections happened at home, how could highly vulnerable people 'shield' at home often in multigenerational homes?

<https://www.medrxiv.org/content/10.1101/2020.04.04.20053058v1>

Countries that have successfully managed COVID-19 such as South Korea and Taiwan (the UK has the highest death rate in the world from COVID-19 based on the data recently) have not pursued 'shielding', they have relied on evidence-based epidemiological strategies such as an effective test and trace programme.

3. The third issue is that 'shielding' is completely impractical. From anecdotal reports, patients were ignoring the guidance, in some cases it was impossible to adhere to the rules. Most older and vulnerable people rely on care from social and community services, 65% rely exclusively on care from family and friends (<https://www.ioaging.org/aging-in-america>). How can you shield when you are dependent on such a level? Highly vulnerable people also need a lot of contact with medical and other services. Care

	homes where a large number of clinically vulnerable patients live have been a hotbed of COVID-19 infection and very high mortality both the in the UK and Sweden, showing the inadequacy of the whole 'shielding' strategy. Shielding is a political and economic programme with no evidence, therefore identifying people in the first place from computer records to 'shield' is not clinically useful information in my opinion. All these must be acknowledged clearly by authors in the article.
--	---

REVIEWER	James Galloway King's College London
REVIEW RETURNED	09-Jul-2020

GENERAL COMMENTS	This is an interesting manuscript. The population management angle is novel in the covid-19 setting, and I thought this was a fascinating approach that will influence other health providers. The premise of the work is very UK centric, although the introduction and methods articulated well the concept of shielding. The methodology uses a machine learning approach, with k-prototype clustering. I have not used this method myself, so speak with limited insight and cannot verify exactly what has been done. However, I understand the approach, and conceptionally it makes sense. The results tables are clear and logical. The figures, especially the geospatial mapping, are very informative. The most controversial aspect of the work is the formation of the six clusters. These are very interesting, and I suspect the discussion needs to capture some additional caveats. The mental health cluster might be misleading... it would be important to know what proportion of these were dementia? It is a small group, and I am suspicious that 'mental health' might be the wrong title for what is mainly dementia. The drug monitoring cluster is important to consider as well. It needs some context around which drugs need monitoring in the discussion. One would anticipate that immune suppression would make up a majority of this cohort. However, many other drugs require monitoring - and understanding this definition more clearly would be helpful. Finally - in terms of transparency, I appreciate the raw data cannot be made public, however the table shells for the raw data, the data variable keys and also the analysis code from python etc. could be posted in GitHub. This would all be useful to any other region looking to replicate the work.
---

VERSION 1 – AUTHOR RESPONSE

We thank the reviewers for their helpful and insightful comments on our manuscript. We have made the suggested revisions and additions with our responses detailed below. We have also uploaded the R code we implemented for the k-prototypes analysis to our GitHub page as per reviewer comments.

Reviewer: 1

Please state any competing interests or state 'None declared':

I was involved in drawing up the shielding criteria for rheumatology patients

Please leave your comments for the authors below This interesting paper describes the use of Population Health management methods to identify and characterise individuals requiring shielding and their health care needs The paper describes the processes used to identify the individuals and then cohorts them into 6 clusters. They have identified 6 distinct cohorts. It is interesting to note that 4 of the 6 had a high proportion of females and 2 of the 6 had a median age below 60 years. They discuss the fact that the shielding criteria failed to take in to account factors such as age, ethnicity and gender that have since emerged as predictors of poor outcome.

---**Author response:** We thank the reviewer for these positive comments – we too found it interesting that 4 of the 6 clusters had a higher proportion of women and subsequently discuss this with regards to known demographics (e.g. higher rate of asthma/autoimmune diseases) under section 4.3 of the Discussion.

What would be really interesting would be to know how many of these patients developed COVID-19 and what their outcomes were. Is this possible? Knowing which of the 6 clusters was actually at highest risk would allow further targeting of resources and inform future shielding decisions.

---We agree with the reviewer that this is an important further question and are currently in the process of acquiring and validating the data that is necessary to do this in our population of 1 million people. Because this will be a substantial piece of work in its own right, that we anticipate will also have important lessons for other PHM units, we propose to publish these findings separately when we are confident we have fully and specifically ascertained deaths from COVID-19 (the reviewer will be aware of the recent concerns that have been raised about how death from COVID-19 is recorded in government statistics).

We have included the following text under 'Future work':

'Two initial priorities for further work are to apply these clusters to other geographic settings to determine whether the clusters hold in other healthcare settings and to prospectively determine the actual risk of severe COVID-19 illness and subsequent outcomes in people who are determined to be at high-risk. With planned developments of our dataset, a further study could examine in greater depth the observed risks and outcomes in terms of hospitalisation, discharges to care settings and mortality among the shielded population'

We provide the reviewer with some preliminary explorations of GP suspected and GP confirmed COVID-19 cases recorded in EMIS collected from February to May 2020. This is work in progress and we acknowledge several caveats (further validation required, potential recording bias, relatively low infection rates in SW), hence not including this in the current paper.

Cluster	Proportion of GP suspected COVID-19	Proportion of GP confirmed COVID-19	Proportion in a care home
1	9.41	0.59	6.47
2	7.22	0.80	7.14
3	3.40	0.15	0.13
4	1.68	0.17	0.59
5	2.02	0.12	1.54

6	2.57	0.15	1.80
---	------	------	------

Table R1: Proportion of GP suspected COVID 19, GP confirmed COVID-19 and care home residents per cluster recorded from February 2020 to end of May 2020. The GP confirmed field is based on a GP or practice admin member coding it in EMIS because it was on a discharge summary or other source and GP suspected is based on GP assessment of the patient.

Cluster 1 (complex mental health) and 2 (older complex) had the highest proportions of people with suspected cases; 9.4% and 7.2% respectively, and confirmed cases; 0.6% and 0.8% respectively. The proportion of each remaining cluster with a suspected or confirmed case was less than 3.4% and 0.2% respectively. These differences may signal differences in clinical vulnerability. They might also be explained by differences in acquisition of the virus and/or testing practices. Both of these could be related to institutionalisation with 6.5% and 7.1% of clusters 1 and 2 respectively being care home residents.

Minor corrections:

Page 3, line 40 – missing end bracket

---Author response: Thank you for bringing this to our attention, we have made the corresponding amendment.

Reviewer: 2

Please leave your comments for the authors below Thank you for the opportunity to review. I think the study is well done from a methodological point of view but I have very strong reservations about the study's clinical utility or purpose. I am pleased that the authors have discussed some of the problems in the limitations of the study. PHM methods involve a lot of work and effort as the authors can tell. The authors have done a good job of analyzing a huge amount of data which is commendable.

---Author response:

Thank you for recognising the methodological strengths of the paper. Whilst acknowledging the limited direct clinical impact of the paper, we have recognised this in the manuscript and provided a clear PHM focus for the paper, because we believe our findings have important implications for health systems, commissioners and policy makers. They will also be of interest and generate discussion amongst clinicians. This has influenced our choice of BMJ Open rather than a traditional entirely clinical journal.

I have briefly summarised my issues with the study below.

1. The limitations show that PHM is not a reliable way to identify high risk individuals. There are problems with information bias - incorrect/incomplete/inadequate coding, relying on error-prone prescriptions datasets etc. These limitations clearly show the problems of the approach adopted in the UK. Identifying people through PHM is probably better than the way it was done by the NHS England but this is still a centralised computer generated list. Many clinicians in primary care including myself found the classification of most of our 'highly vulnerable' patients incorrect, those who were supposed to 'shield' did not receive any information but many patients were incorrectly classified as very high risk based on outdated or incorrect data held centrally. A centralised, top down approach of identifying high risk individuals is ineffective and certainly leads to confusion, frustration, misallocation of resources and potential harm. A better approach is supporting and providing resources to practices and community care facilities to identify high risk individuals using their knowledge of individual patients under their care and their health needs.

---Author Response:

We share the experience of the challenges in implementing the shielding process in the NHS with our own patients. However, local policy makers and health leaders still need methods to manage the health of their population (rather than at individual practice level) accounting for central policy. Our paper shows how PHM segmentation approaches can be used to identify geographic areas that may be high risk (acknowledging the limitations of the data), and support a health system in the response to a pandemic.

We agree that GP practices and other community care facilities ultimately hold responsibility for making decisions for those who require shielding and have included the following text under Section 4.5 of the Discussion.

'...In addition, research is needed to understand and maximise the classification accuracy of search criteria used for population level datasets, as has been performed in the clinical practice research database,⁴⁹ and how these can be integrated as support mechanisms for front-line clinicians, who ultimately hold responsibility for making decisions on who meets the criteria for shielding. Critically, it needs to be established whether shielding is effective.'

We have amended the Section 4.2 of the Discussion to include the following:

'Finally, a computer driven approach can identify individuals quickly from large patient lists and records but additional benefit could be obtained by providing further support to practices to identify high risk individuals using their knowledge of individual patient health and care needs.'

2. As the authors acknowledge there is absolutely no evidence that 'shielding' works at all. The problem is the assumptions about 'shielding' which was a political strategy adopted in the UK, Sweden and some other Western countries. It is not a scientific strategy. 'Shielding' is based on 2 erroneous assumptions. 1) If only a small proportion of the population 10-15% are very highly vulnerable to COVID-19 (and economically not productive), they could be easily isolated from 90% of the population who will likely have a mild illness and will develop 'herd immunity'. The Chief Scientific Advisor on 12 March 2020 said: "Our aim is not to stop everyone getting it, you can't do that. And it's not desirable, because you want to get some immunity in the population. We need to have immunity to protect ourselves from this in the future." The underlying objective was when initially announced, to 'cocoon' those 'most at risk' whilst leaving the economy open outside, in the hope that those who survive will develop 'herd immunity'. There is no evidence that those infected develop immunity and it quickly became apparent that the strategy would lead to hundreds of thousands of deaths. A nationwide lockdown was introduced shortly after. Furthermore, evidence from China shows that 80% of infections happened at home, how could highly vulnerable people 'shield' at home often in multigenerational homes? <https://www.medrxiv.org/content/10.1101/2020.04.04.20053058v1> Countries that have successfully managed COVID-19 such as South Korea and Taiwan (the UK has the highest death rate in the world from COVID-19 based on the data recently) have not pursued 'shielding', they have relied on evidence-based epidemiological strategies such as an effective test and trace programme.

---Author response:

As the reviewer recognises, we have acknowledged the lack of evidence for shielding policy in our paper. We would agree with the reviewer that shielding and necessarily perhaps much of the response to COVID-19 can be framed as a political question, and has important practical implications.

We have made several changes to the discussion to address the reviewers' comments in further detail and to highlight the problems with shielding policy. These changes include the following:

Under Section 4.5

'...Shielding may be undermined by non-concordance with the rules, a reliance on social, medical and care interactions, or from being resident in a high risk setting such as a care home. This is especially important given the lack of evidence for the rest of the non-shielded population having acquired 'herd immunity'. Without such evidence there is a risk to the wellbeing of current and future shielded populations, induced by the shielding process. Finally, it should be examined whether there are more effective strategies such as widespread use of face coverings in public and effective 'track and trace' systems.'

Under Section 4.4

'There are also further limitations with the principle of shielding. Firstly, shielding has a very limited evidence base, and therefore, as with much of the policy response to COVID-19, the international implementation has been varied. For example, South Korea did not implement shielding and has a

very low number of deaths reported from COVID-19. However, at this stage we believe that limitations in the comparability of the death rate (or indeed excess deaths) between international states preclude the comparison of death rates by interventions on a state-by-state basis. Secondly, shielding can be difficult to implement on a practical level for the individual. Further work could explore the strategies that individuals used to operationalise shielding on an everyday basis, the extent to which this adhered to national guidelines, and whether there were any barriers to implementation that could have been better mitigated by a more comprehensive response from the health system. However, regardless of these limitations our report demonstrates how PHM methods can be applied at pace, during a pandemic, to identify geographic areas and characterise clusters of people who likely have additional health needs and support actionable insights to safeguard population health.'

'...It is important to note that at the time of writing it is not known whether the UK shielding strategy has been effective in terms of reducing the burden of COVID-19 or overall harm for that defined population. Our findings are therefore relevant to healthcare planners and clinicians because they offer insights at the level of the population...'

3. The third issue is that 'shielding' is completely impractical. From anecdotal reports, patients were ignoring the guidance, in some cases it was impossible to adhere to the rules. Most older and vulnerable people rely on care from social and community services, 65% rely exclusively on care from family and friends (<https://www.ioaging.org/aging-in-america>). How can you shield when you are dependent on such a level? Highly vulnerable people also need a lot of contact with medical and other services. Care homes where a large number of clinically vulnerable patients live have been a hotbed of COVID-19 infection and very high mortality both the in the UK and Sweden, showing the inadequacy of the whole 'shielding' strategy.

Shielding is a political and economic programme with no evidence, therefore identifying people in the first place from computer records to 'shield' is not clinically useful information in my opinion. All these must be acknowledged clearly by authors in the article.

---Author Response:

We share the reviewers' concerns that shielding has important practical implications for individuals. We have addressed these issues in further detail:

Section 4.4 (as included in above response to Reviewer but also relevant to address the third concerns)

'Secondly, shielding can be difficult to implement on a practical level for the individual. Further work could explore the strategies that individuals used to operationalise shielding on an everyday basis, the extent to which this adhered to national guidelines, and whether there were any barriers to implementation that could have been better mitigated by a more comprehensive response from the health system. However, regardless of these limitations our report demonstrates how PHM methods can be applied at pace, during a pandemic, to identify geographic areas and characterise clusters of people who likely have additional health needs and support actionable insights to safeguard population health.'

Section 4.5 (as included in above response to Reviewer but also relevant to address the third concerns)

'...In addition, research is needed to understand and maximise the classification accuracy of search criteria used for population level datasets, as has been performed in the clinical practice research database,⁴⁹ and how these can be integrated as support mechanisms for front-line clinicians, who ultimately hold responsibility for making decisions on who meets the criteria for shielding. Critically, it needs to be established whether shielding is effective. Shielding may be undermined by non-concordance with the rules, a reliance on social, medical and care interactions, or from being resident in a high risk setting such as a care home. This is especially important given the lack of evidence for the rest of the non-shielded population having acquired 'herd immunity'. Without such evidence there is a risk to the wellbeing of current and future shielded populations, induced by the shielding process. Finally, it should be examined whether there are more effective strategies such as widespread use of face coverings in public and effective 'track and trace' systems.'

Part of the research was as the reviewer says to identify people from computer records who require shielding, however we take this further and characterise the shielded group in more detail using a wide set of attributes. For example, we believe it is of value to identify geographic areas where people who are likely to be at high risk live, and to identify clusters of people who are likely to fall within the shielding criteria. Our initial analyses for instance showed where potential demand was mismatched with local GP capacity and resilience plans were made to ensure continuity of service if certain areas became overwhelmed. Local health leaders and policy makers need to identify areas of greatest need, and characterise the people who are identified, by national policy makers, as being at highest risk, in order to attempt to plan and mitigate the additional risk to these people and support the health of the population.

Reviewer: 3

Please leave your comments for the authors below This is an interesting manuscript. The population management angle is novel in the covid-19 setting, and I thought this was a fascinating approach that will influence other health providers. The premise of the work is very UK centric, although the introduction and methods articulated well the concept of shielding.

---Author response: We thank the reviewer for these positive comments on the manuscript. We agree that the approach taken in the study is UK centric, because of this we endeavoured to include detailed methods and definitions in order for other local authorities to carry out similar analyses and to consider the methods further afield. We have altered the wording slightly in the discussion to reflect this.

Under Section 4.2

'The study had its limitations. Firstly, the findings were derived from a single health and care system in one country (UK) which may hinder generalisability. However, the BNSSG population closely matches the age and sex distribution for England using ONS data and PHE data.^{32 33} The results may therefore be portable to other English systems, or countries with similar demographics and health status.'

With the addition of including the R code and data shells as the reviewer suggests below, this also enhances the transferability of the research.

The methodology uses a machine learning approach, with k-prototype clustering. I have not used this method myself, so speak with limited insight and cannot verify exactly what has been done. However, I understand the approach, and conceptionally it makes sense.

---Author response: We aimed to include a broad description of the cluster analysis in the main manuscript and included all the additional details of the approach in Supplementary Material C for those who wanted to delve further into the methodology. The authors have expertise in population segmentation and have used clustering approaches for several different projects.

The results tables are clear and logical. The figures, especially the geospatial mapping, are very informative.

The most controversial aspect of the work is the formation of the six clusters. These are very interesting, and I suspect the discussion needs to capture some additional caveats.

The mental health cluster might be misleading... it would be important to know what proportion of these were dementia? It is a small group, and I am suspicious that 'mental health' might be the wrong title for what is mainly dementia.

---Author response: We have added more detail behind the composition of Cluster 1; the complex mental health group, as we agree that it is a complex group made up different types of individuals and further granularity is needed. This group had the highest proportion of people with dementia 10.59% indicated in Table 2, followed by Cluster 2 with 8.16%. Half the individuals with dementia had a mental health condition, whereas 17.76% of individuals without dementia had a mental health condition. We investigated different numbers of clusters (6,7,8,9 – Supplementary Material C), and the complex mental health group (around 170 people) was consistently identified with around 80% of

the individuals in this group having a mental health condition. Interestingly, only 10.59% had dementia, membership of this group was therefore mainly attributed to the 78.82% of individuals having a mental health condition. We have updated the text with the following to gain additional insight into this group.

Under Section 3 (Cluster Analyses Results)

‘Cluster 1 contains only 0.57% of the high-risk group and accounts for those with the highest need in regard of mental health services (79% had a mental health diagnosis). This cluster also had the greatest proportion of individuals with dementia (10.59%). Those with dementia in this cluster had a mean age of 78 versus 56 for those without dementia. Half the individuals with dementia had a mental health condition, whereas 17.76% of individuals without dementia had a mental health condition. Although this cluster had the highest proportion of people with dementia, membership of this group was largely defined by 78.82% of individuals having a mental health condition. Median Charlson comorbidity index for those with dementia was 6 (IQR: 5-7.75), those without dementia was 3 (IQR: 2-4).’

The drug monitoring cluster is important to consider as well. It needs some context around which drugs need monitoring in the discussion. One would anticipate that immune suppression would make up a majority of this cohort. However, many other drugs require monitoring - and understanding this definition more clearly would be helpful.

---Author response: We agree that this aspect needs further discussion and specification.

We have included under Table 1 and 2 the following text to provide further information on the drugs which require monitoring:

**‘Drugs that require monitoring include immunosuppressant/immunomodulators (IMMIMM) (previous 6 months), biologic/monoclonal (IMMBIO) (previous 6 months) including rituximab (previous 12 months) and other drugs requiring monitoring (previous 2 months), fully specified in Supplementary Material B’*

We have also included in Supplementary Material B all the individual drugs included in the data search which make up IMMIMM, IMMBIO and other drugs which require monitoring for full transparency. We have clarified the drugs included for this variable in Table C.3 (Supplementary Material C) which details the clustering and non-clustering variables used for analyses and how they were operationalised.

Finally, we have included the following detail under Section 3 (Cluster Analyses Results):

‘Cluster 5 is characterised by the majority (78%) requiring physiological, biochemical or pharmacological monitoring of prescribed medication. Of this cluster, 60% had an immunosuppressant/immunomodulator prescribed in the previous 6 months, 22% had a biologic/monoclonal medication prescribed in the previous 6 months and 11% had another type of drug requiring monitoring in the previous 2 months (e.g. lithium, warfarin, anti-psychotic medications - see Supplementary Material B and C).’

	Immunosuppressant/ immunomodulator IMMIMM previous 6 months	Biologic/Monoclonal IMMBIO previous 6 months (including rituximab previous 12 months)	‘Other Drugs’ Requiring Monitoring previous 2 months
Drug Monitoring Cluster (6892)	4135 (60.00%)	1516 (22.00%)	764 (11.10%)

Finally - in terms of transparency, I appreciate the raw data cannot be made public, however the table shells for the raw data, the data variable keys and also the analysis code from python etc. could be posted in GitHub. This would all be useful to any other region looking to replicate the work.

---**Author response:** We think this suggestion will complement the content of the Supplementary Material B where we specify variable definitions and coding lists and will enable reproducible research. We have added the data shells and R code for analyses onto the BNSSG Modelling and Analytics GitHub page [<https://github.com/nhs-bnssg-analytics/>], [<https://github.com/nhs-bnssg-analytics/kprotocluster>].

VERSION 2 – REVIEW

REVIEWER	Elizabeth Price Great Western Hospital Swindon UK Involved in developing shielding criteria for rheumatology patients
REVIEW RETURNED	20-Aug-2020

GENERAL COMMENTS	The authors have addressed the points I raised in my original review
--

REVIEWER	Dr Mohammad S Razai Population Health Research Institute, St George's University of London, UK
REVIEW RETURNED	12-Aug-2020

GENERAL COMMENTS	Thank you for the opportunity to review this manuscript again. I believe the revisions have improved the paper and I congratulate the authors for their work on this paper. I think the response to my queries and concerns are adequately addressed. I am happy to recommend publication. The paper's methodology and some of the issues raised
--

REVIEWER	James Galloway King's College London
REVIEW RETURNED	19-Aug-2020

GENERAL COMMENTS	I read with interest the authors' responses to all the comments. They have taken time to carefully address all the issues. The paper is a useful and important addition to the evidence base.
---